# The Detection of Pulp Stones with Automatic Deep Learning in Panoramic Radiographies: An AI Pilot Study [note 1]

**DOI:** 10.3390/diagnostics14090890

**Published:** 2024-04-24

**Authors:** Ali Altındağ, Serkan Bahrilli, Özer Çelik, İbrahim Şevki Bayrakdar, Kaan Orhan

**Affiliations:** 1Department of Oral and Maxillofacial Radiology, Faculty of Dentistry, Necmettin Erbakan University, 42090 Konya, Turkey; serkanbahrilli@gmail.com; 2Department of Mathematics and Computer Science, Faculty of Science, Eskişehir Osmangazi University, 26480 Eskişehir, Turkey; ozercelik05@gmail.com; 3Department of Oral and Maxillofacial Radiology, Faculty of Dentistry, Eskişehir Osmangazi University, 26040 Eskişehir, Turkey; ibrahimsevkibayrakdar@gmail.com; 4Department of Oral and Maxillofacial Radiology, Faculty of Dentistry, Ankara University, 06560 Ankara, Turkey; call53@yahoo.com

**Keywords:** artificial intelligence, deep learning, panoramic, pulp stone, YOLOv5

## Abstract

This study aims to evaluate the effectiveness of employing a deep learning approach for the automated detection of pulp stones in panoramic imaging. A comprehensive dataset comprising 2409 panoramic radiography images (7564 labels) underwent labeling using the CranioCatch labeling program, developed in Eskişehir, Turkey. The dataset was stratified into three distinct subsets: training (*n* = 1929, 80% of the total), validation (*n* = 240, 10% of the total), and test (*n* = 240, 10% of the total) sets. To optimize the visual clarity of labeled regions, a 3 × 3 clash operation was applied to the images. The YOLOv5 architecture was employed for artificial intelligence modeling, yielding F1, sensitivity, and precision metrics of 0.7892, 0.8026, and 0.7762, respectively, during the evaluation of the test dataset. Among deep learning-based artificial intelligence algorithms applied to panoramic radiographs, the use of numerical identification for the detection of pulp stones has achieved remarkable success. It is expected that the success rates of training models will increase by using datasets consisting of a larger number of images. The use of artificial intelligence-supported clinical decision support system software has the potential to increase the efficiency and effectiveness of dentists.

## 1. Introduction

Pulp stones, characterized as calcified masses within the dental pulp, may manifest across various dental conditions including permanent and primary dentition and erupted and unerupted teeth, as well as in both healthy and compromised dental structures. Their occurrence can be localized to a single tooth or extend across the entire dentition [1]. While the precise etiology of pulp calcifications remains elusive, their development is thought to involve a complex interplay of various factors. These include interactions between epithelial and pulp tissues, disturbances in pulp circulation, tissue degeneration, periodontal pathology, dental caries, orthodontic interventions, and chronic inflammatory processes, as well as demographic factors such as age, gender, genetic predisposition, and idiopathic influences [2,3].

Pulp calcification is detected through radiographic and histological examinations. However, for calcification to be discernible in radiographs, it needs to possess a certain size (>200 µm) and degree of mineralization [4,5]. Although they have some limitations, it is a known fact that radiological methods are the only technique that can detect calcification clinically and non-invasively [6].

The currently accepted clinical view is that pulp stones are of no importance other than the possibility of causing difficulties during endodontic treatment, such as making canal positioning difficult and preventing access to the canal [7]. However, it is known that pulp stones affect the response to pulp tests, causing tooth perforation, instrument fracture, and even tooth loss [8]. The extraction of pulp stones from the pulp chamber presents a complex and intricate procedure, demanding a high level of proficiency, precision, and expertise. Additionally, it necessitates the utilization of specialized magnification tools and appropriate equipment. Therefore, it may be necessary to refer to an endodontist when pulp stones are detected [9].

Machine learning represents a subset of artificial intelligence, enabling computers to execute tasks without explicit programming, but rather through the analysis of extant data relationships. Conversely, deep learning pertains to a domain that includes machine learning algorithms, specifically those that involve artificial neural networks characterized by one or more hidden layers [10]. Recently, the advancements in the realm of dental digital imaging have paved the way for the inception of software architectures grounded in deep learning, facilitating the identification of dental caries, periodontal ailments, and dental abscesses from radiographic images [11].

Recognizing objects within images or videos poses a formidable challenge for computers compared to the innate proficiency of humans in this cognitive task. The advent of deep learning, particularly leveraging artificial neural networks, has garnered considerable attention, especially in tandem with the expanding availability of labeled data. A focal point of this technological progression is object recognition, a process integral to artificial intelligence applications.

Object detection, a critical facet of image processing, involves precisely locating specific objects within an image or video feed. Numerous algorithms have been devised for this purpose, with deep learning methodologies, particularly those based on convolutional neural networks (CNNs), emerging as pre-eminent and widely adopted tools in contemporary research and applications [12]. The efficacy of CNNs in object detection stems from their adeptness in discerning hierarchical features through a multilayered architecture.

The fundamental operational paradigm of a CNN encompasses one or more convolutional layers, with subsequent subsampling layers, and culminates in one or more fully connected layer, akin to a conventional multilayer neural network [13]. In this context, the mathematical convolution process mirrors a neuron’s responsiveness to stimuli within its designated receptive field. The amalgamation of these components endows CNNs with a robust capacity for discerning and localizing objects within complex visual data.

The object detection methodology proves applicable across various domains, including classification and human face recognition. Object detection algorithms, with a focus on zone recommendation, encompass prominent models such as R-CNN, Mask-RCNN, R-FCN, SPP-net, FPN, Fast R-CNN, and Faster R-CNN, and include YOLOv5 [14].

The YOLO (You Only Look Once) algorithm is a convolutional neural network (CNN)-based deep learning model designed for the prediction of object bounding box coordinates, associated probability values, and class categorization within images. Renowned for its commendable performance in both speed and accuracy, the YOLO algorithm is particularly suited for real-time object detection applications [12].

In this study, unlike previous research studies, new effort was made in the field of automatic pulp stone detection. While in previous studies, Mask R-CNN and YOLOv4 architectures were primarily used to detect pulp stones in bitewing images, in this research, a new approach was presented, and the effectiveness of YOLOv5, a current architecture, was evaluated in the field of panoramic images. This departure from other methodologies underscores our commitment to advancing the field by exploring innovative means of accurate and effective pulp stone detection. In this study, where YOLOv5’s architecture is used in the context of panoramic radiographs, it is thought to not only expand the scope of automatic detection techniques but also provide new and different information to the existing literature on dental radiology.

## 2. Material and Methods

### 2.1. Ethical Approval

The research protocol obtained ethical clearance from the Research Ethics Committee of the Faculty of Dentistry at Necmettin Erbakan University, adhering to the principles outlined in the Helsinki Declaration of Human Rights (Approval Number: 2022/17-136; Approval Date: 28 April 2022).

### 2.2. Study Sample

The panoramic images used in this research were obtained from people who applied to the clinic due to various dental disorders between January 2020 and September 2021. These images were captured utilizing the Planmeca Promax 2D Panoramic system (Planmeca, Helsinki, Finland; at a 68 kVp, 14 mA, and 12 s). The acquisition protocols adhered to the manufacturer’s stipulated guidelines for respective imaging system. Notably, prior to their integration into the study dataset, all images underwent an anonymization process.

Panoramic images meeting diagnostically acceptable criteria and surpassing the 16-year age threshold, devoid of prevalent bone or dental pathologies, were incorporated into the study dataset. Conversely, panoramic images characterized by suboptimal quality or the presence of artifacts or depicting individuals with specific dental disorders (such as dentinogenesis imperfecta, dentin dysplasia, etc.) were systematically excluded from consideration. Additionally, panoramic images capturing subjects under the age of 16 were not included in the study parameters. However, when working with real-world data, no diagnostic quality conditions were taken into account, and all panoramic images that met the above-mentioned criteria were included in the study.

#### 2.2.1. Labeling

The examinations were conducted by two oral and maxillofacial radiologists denoted as A.A. and S.B. For the purpose of assessing inter-observer agreement, Cohen’s kappa statistics were employed, utilizing a subset comprising 20% of the total images. The delineation of dental regions wherein pulp stones were identified, namely incisors, canines, premolars, and molars, was executed with the CranioCatch v1.5 program (CranioCatch, Eskişehir, Turkey).

#### 2.2.2. Training

Within the scope of this study, the YOLOv5 architecture (Figure 1), recognized as a deep learning-based image segmentation model, was adopted. The model’s performance was evaluated utilizing the confusion matrix, a metric adept at visually representing system predictions against actual scenarios, thereby facilitating a comprehensive evaluation of machine learning algorithm efficacy.

In total, 7564 labels were made on 2409 panoramic images. The dataset was partitioned into three distinct groups; the training group, comprising 1929 samples with a total of 6033 labels; the validation group, consisting of 240 samples with 766 labels; and the test group, also comprising 240 samples with 765 labels.

The 2409 images came in different sizes. For this reason, the images were resized to 1024 × 512. The proposed artificial intelligence (AI) model, denoted as CranioCatch (Eskişehir, Turkey), employs a deep convolutional neural network (CNN) strategy for pulp stone (PS) detection. The model was specifically trained by utilizing 500 epochs with the Pytorch architecture, incorporating the COCO dataset and a learning rate of 0.01. The model is illustrated in Figure 2. Notably, the identification of PS necessitated the utilization of a distinct deep CNN. The training regimen involved 7000 steps, executed on a computer system (PowerEdge T640 Compute Server (Dell Inc., Austin, TX, USA), PowerEdge T640 GPU Compute Server (Dell Inc., Austin, TX, USA), PowerEdge R540 Storage Server (Dell Inc., Austin, TX, USA) in the computer equipment of the Eskişehir Osmangazi University, Faculty of Dentistry-AI Laboratory) equipped with 16 GB RAM and NVIDIA GeForce GTX 1660 TI (NVIDIA, Santa Clara, CA, USA). The training and validation datasets were instrumental in predicting and optimizing the weight factors of the CNN algorithm.

#### 2.2.3. Success Evaluation

The assessment of the model’s efficacy was performed using a confusion matrix, a significant table commonly employed to assess system performance by summarizing predicted and actual scenarios. The principal metrics utilized for assessing model performance include true positive (TP), denoting PSs correctly identified by both experts and AI; false negative (FN), indicating PSs identified by experts but overlooked by AI; and false positive (FP), signifying PSs not identified by experts but flagged by AI.

Following the computation of TP, FN, and FP, the subsequent metrics were derived: sensitivity (recall), calculated as TP/(TP + FN); precision, computed as TP/(TP + FP); and F1 score, determined by 2TP/(2TP + FP + FN). These metrics collectively serve as robust indicators for assessing the performance and efficacy of the AI model in the context of PS detection.

## 3. Results

The kappa value of intra-observer agreement for A.A. and S.B. was found to be 0.946 and 0.913, respectively. The inter-observer agreement value was 0.921. Pulp stones were detected in the panoramic images of 2409 patients. Upon scrutiny of the radiographs in which the algorithm failed to detect or exhibited errors, it was noted that the YOLOv5 architecture demonstrated inadequacy in identifying small pulp stones located within the incisor, canine, and premolar teeth, attributed to issues such as superpositions and artifacts. The diminished visibility of root pulp in comparison to crown pulp contributes to the challenges encountered in detecting and accurately identifying pulp stones. The relatively smaller size of the dataset pertaining to root pulp further exacerbates this issue, leading to instances of missed or erroneous detection within this particular region (Figure 3). The sensitivity, precision, and F1 results obtained using the YOLOv5 architecture in the test dataset were found to be 0.8026, 0.7762, and 0.7892, respectively (Table 1).

Figure 4 provides a visual representation of the training results obtained from YOLOv5, demonstrating variations in key metrics including objectness loss, box loss, segmentation loss, classification loss, recall, precision, and mean average precision across 500 epochs for both training and validation datasets. Box loss delineates spatial centrality and extent, while objectness measures the likelihood of parameter occupancy. Segmentation and classification losses indicate algorithmic predictive efficacy. Model performance, characterized by recall, precision, and mean average precision, evolves throughout training, with peak-performing models highlighted in red for each parameter.

Figure 5 utilizes correlograms, specialized 2D histograms, for the simultaneous visualization of multi-axial data, providing an immediate overview of dataset relationships and label distributions. This graphical representation integrates the position coordinates, width, and height of labels related to key parameters. The correlations among these labels are visually compared for a nuanced understanding.

## 4. Discussion

The non-invasiveness and routine integration of radiological evaluation in dental examinations make it a very advantageous diagnostic tool. Various radiographic methods have been used for the diagnosis of pulp stones (PSs) in many studies. For example, Tamse et al. [15] conducted evaluations using both periapical and bitewing radiographs to detect PSs and examined the discrepancy between these two methods. However, the limitation of bitewing and periapical radiographs is that they cannot evaluate all teeth in the jaw simultaneously. Although CBCT stands out as the most suitable imaging modality for pulp stone detection due to its ability to reduce superpositions and provide superior resolution, its routine use is limited by the associated high radiation dose. CBCT is generally reserved for specific indications rather than being a standard component of routine radiographic examinations. Considering these factors, and consistent with previous research findings, panoramic radiographs that show the entire jaws and are routinely used in clinical practice were selected to be applied in this study to detect the presence of PS.

The incorporation of artificial intelligence (AI) within the medical domain, propelled by advancements in deep learning and neural methodologies, has permeated into the field of dentistry [14,16]. Significantly, AI exhibits promising accuracy in the interpretation of medical imaging modalities, including X-rays, Cone-Beam Computed Tomography (CBCT), Computed Tomography (CT) scans, and Magnetic Resonance Imaging (MRI). Applications of machine learning (ML) and deep learning (DL) in dental practice hold substantial potential, presenting advantages for clinicians and indicating prospective integration into the dentist’s armamentarium [14]. The broader domain of AI in healthcare encompasses disease risk prediction, the identification of genetic anomalies, disease diagnosis, prognostic evaluation, and scrutiny of anatomical structures. In dentistry, AI technologies primarily serve as guiding tools for clinicians, enhancing clinics’ operational efficiency and supporting overall clinic systems. Recent investigations underscore the escalating utilization of deep learning methodologies, particularly the proficiency of convolutional neural networks, renowned for their adept recognition of repetitive image patterns [17,18].

Pulpal calcifications represent densely calcified masses that can manifest within the pulp tissue, occurring ubiquitously and varying in size [19,20]. These calcifications are broadly categorized into two classes: pulp stones and dystrophic calcifications [20,21]. The nomenclature for these entities, including terms such as PSs, pulpal calcification, nodules, denticles, or dystrophic calcification, is frequently utilized interchangeably, alluding to calcified masses within the pulp chamber [20,22]. Previous investigations concerning pulpal calcifications have commonly aimed to identify calcifications discernible from the dentinal wall, often neglecting an in-depth analysis of their structural characteristics and spatial distribution [2,20,23].

In the present study, a focus was placed on detecting calcifications distinctly discernible solely within the pulp of the teeth, facilitated by the chosen radiographic imaging technique. Subsequently, artificial intelligence learning was applied to this dataset. The principal objective of this investigation was to assess the efficacy of employing DL methodologies for the identification of PS, with a specific focus on panoramic radiography as the imaging modality.

The identification of pulpal calcifications, once uncovered, generally does not necessitate any dental intervention [2,14]. However, if symptoms akin to pain arise in the tooth, endodontic treatment may become imperative. Pulpal calcifications have the potential to impede endodontic procedures or other analogous dental interventions. Calcifications can impede the visibility of canal openings, posing challenges in discerning the canal orifice. Similarly, calcifications within the canal lumen have the potential to completely block the lumen, further complicating endodontic procedures [24]. During dental interventions, inadvertent procedural errors, such as the inadequate attainment of the root canal’s complete working length, the fracture of dental instruments, or perforation, may occur [6,25,26].

Given that artificial intelligence exhibits enhanced proficiency in detecting features challenging for the human eye [27], the anticipation is that the incorporation of AI-assisted methods may assist clinicians in identifying calcifications more effectively [28,29]. The trained deep learning model demonstrated approximately 80% success in detecting the presence of pulpal calcifications. It is envisaged that the success rate of our deep learning model will see improvements with the utilization of larger datasets.

The objective of artificial intelligence (AI) investigations in dental radiology is to enhance the interpretation of routine radiographs, facilitating expeditious analysis and decision-making for complex cases. Furthermore, AI endeavors to provide support for novice dentists in diagnostic processes. Image segmentation, a pixel-level classification task, involves grouping image elements pertaining to the same object class [30]. Frequently utilized in medical applications to delineate tumor boundaries or quantify tissue volumes, image segmentation models based on deep learning consistently demonstrate heightened accuracy rates, signaling a transformative shift in the domain [14,30].

The R-CNN architecture, initially devised for region-based image detection tasks, underwent iterative enhancements culminating in Faster R-CNN, forming the foundation for Mask R-CNN. Recognized as state of the art in image and instance segmentation, Mask R-CNN, as evaluated by He et al. [31], surpassed preceding models by leveraging RoIAalign, multitasking training, and ResNeXt-101. Its superiority lies in pixel-to-pixel alignment, expediting experimentation, ensuring a rapid system, and facilitating the recognition of round-shaped pixels. This attribute proves particularly advantageous in detecting pulp stones, typically those that are oval in contour, through deep learning methods [14,31].

The Mask R-CNN architecture finds application in various dental domains, notably in dentistry’s caries detection and tooth numbering/segmentation. Moutselos et al. [32] employed Mask R-CNN to detect occlusal caries in 88 intraoral radiographs, utilizing ICDAS scoring. The recall, precision, and F-score values obtained were 0.889, 0.778, and 0.667, respectively. Silva et al. [33] conducted an exhaustive literature review on segmentation methods in dental imaging, reporting Mask R-CNN’s performance metrics. The architecture demonstrated an average accuracy of 92%, specificity of 96%, precision of 84%, recall of 76%, and F-score of 79%, indicative of low false positives and negatives. While direct comparisons with other unsupervised methods may be challenging, these results underscore Mask R-CNN’s superiority.

Within the realm of pulp stone detection, apart from Selmi et al.’s [34] conference statement, there is a notable gap in the literature regarding artificial intelligence-based software utilization. Selmi et al. [34] employed a convolutional neural network (CNN), achieving a correct prediction rate of 76.4% using a Medium Gaussian Support Vector Machine of Residual Network 50 (ResNet-50). Inception v3 attained a 73.1% correct prediction rate with an identical classifier, with ResNet-50 exhibiting a 7% lower false positive rate. In the current study, employing the YOLOv5 architecture yielded approximately 80% sensitivity for pulp stone diagnosis, affirming its superior performance over other object-detection algorithms.

Fariza et al. [35] conducted pulp chamber, dentin, and enamel segmentation on panoramic images, achieving an accuracy of 93.3%. Lee et al. [36] developed a deep learning model targeting proximal caries detection in bitewing radiographs. Their model encompassed the segmentation of various structures including pulp chamber, dentin, enamel, background, and restoration. Performance evaluation using CNNs yielded a precision of 63.29%, recall of 65.02%, and F1-score of 64.14%. Yang et al. [37] compared the efficacy of four distinct deep learning models for tooth segmentation, with a focus on pulp segmentation through the identification of pulp chamber centers. The evaluation of pulp chamber segmentation across 10 CBCT images comprising 512 slices each revealed success rates ranging from 55.90% to 99.78%. Here, our study employed a YOLOv5 deep learning model, achieving an 80% success rate in pulp stone detection.

Another pulp stone detection study in the literature with artificial intelligence is the study carried out by Yüce et al. [38], focusing on bitewing images. The employed algorithm for pulp stone detection in this study was YOLOv4, and the reported success rate of artificial intelligence in this context was documented as 90%. Noteworthy congruence exists between the outcomes of this study and our own. Differences in imaging methods and specific artificial intelligence algorithms used between studies are thought to cause the observed differences in results. This comparison underscores the importance of accounting for methodological divergences when interpreting and contextualizing findings in the domain of artificial intelligence-assisted pulp stone detection.

Salahin et al. [39] conducted a study focusing on the implementation and validation of an automated system for carious lesion detection from smartphone images, employing deep learning techniques. The YOLOv5X and YOLOv5M models demonstrated superior performance compared to alternative models within the same dataset. Similarly, Zhou et al. [40] investigated the efficacy of deep learning methodologies in classifying and detecting recurrent aphthous ulcerations using clinical photographs of the oral region. Their findings indicated that ResNet50 achieved the highest success in image classification, while the pretrained YOLOv5 architecture excelled in object detection. Additionally, Ayan et al. [41] evaluated the caries diagnosis performance of dental students following training with an artificial intelligence application. Using the YOLOv5 program, caries lesions were annotated in 1200 images by two experts, with 1000 randomly selected images utilized for student education and the remaining 200 images reserved for evaluating AI-assisted caries diagnosis. The overall class mean average precision (mAP) scores for YOLOv5s and YOLOv5xs were 63% and 65%, respectively. Additionally, the results of their study showed that students receiving artificial intelligence training in carious lesion detection demonstrated promising results.

Altındağ et al. [14] conducted a study on the detection of pulp stones using artificial intelligence, focusing on bitewing images. In contrast to the approach employed by Yüce et al. [38], they utilized the Mask R-CNN architecture, annotating 1745 pulp stones across 1269 bitewing images. Their investigation yielded precision, sensitivity, and F1 success rates of approximately 90%. Notably, they highlighted the model’s proficiency in detecting pulp stones within the crown region, albeit with a lower success rate in identifying those within the root region. In our study, we opted for the YOLOv5 architecture and panoramic images, distinguishing our research from prior research. Disparities in outcomes may stem from variances in architectural frameworks and imaging modalities. Additionally, differences in pulp stone size between panoramic and bitewing imaging modalities may contribute to observed distinctions. Additional comparative analyses are necessary to further explore and clarify these discrepancies, thus advancing knowledge in this field.

Given the utilization of panoramic images in this study, a heightened level of detail regarding the pulp chambers of the teeth was attainable, facilitated by the expertise of the observers, who were experienced oral radiologists. This proficiency was reflected in the attained high Kappa value, signifying a notable degree of consistency in observations. However, it is imperative to acknowledge that pulpal calcifications present a diagnostic challenge, particularly for dental students with limited perceptual acuity and practitioners, including dentists, who lack specialization in oral radiology and endodontology. The introduction of artificial intelligence assumes a supportive role in mitigating diagnostic challenges for these clinicians and dental students, offering assistance within the clinical workflow, especially for those with limited experiential depth in the nuanced assessment of pulpal calcifications.

### Limitations

This study is beset with several inherent limitations warranting acknowledgment. A notable constraint pertains to the absence of an assessment of the performance exhibited by the participating experts. Another limitation is the use of images obtained from a single panoramic device. Subsequent studies could enhance the robustness of findings by expanding the variety of panoramic devices and incorporating multicenter study designs. The imperative for additional efforts and dedicated research endeavors is underscored, as the meticulous refinement and advancement of the established methodology are essential prerequisites before the prospective practical implementation of the process. This ensures its efficacy as a supportive tool for clinicians and underscores the necessity for a seamless translation of results into clinically meaningful effects.

## 5. Conclusions

Deep learning algorithms have demonstrated efficacy in the detection of pulp stones, thereby offering the potential for software systems supported by artificial intelligence to assist dental practitioners in diagnostic examinations. The YOLOv5 architecture, when employed for pulp stone detection, exhibits approximate sensitivity of 80%. Notably, the accuracy rates within deep learning techniques exhibit an upward trend with the expansion of the dataset. The augmentation of radiographic data in training models correlates with heightened rates of success, underscoring the imperative for increased data volume in future studies aimed at refining and advancing these methodologies.

## Figures and Tables

**Figure 1 diagnostics-14-00890-f001:**
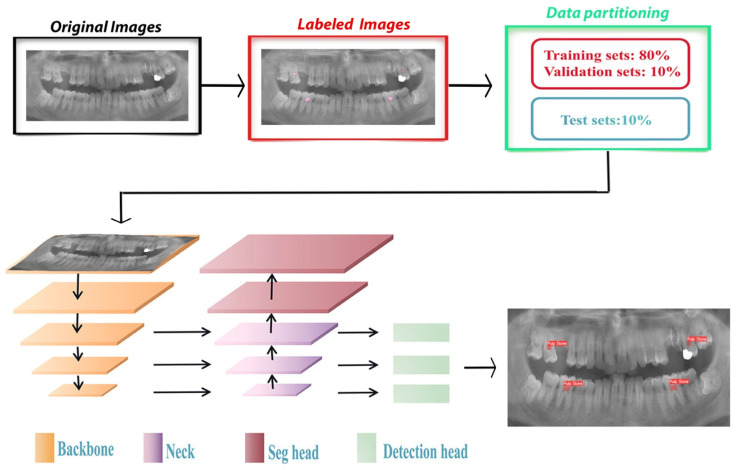
Network structure of YOLOv5.

**Figure 2 diagnostics-14-00890-f002:**
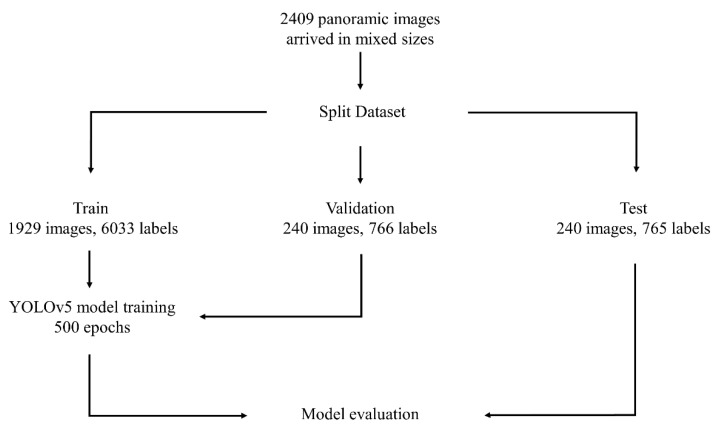
The diagram of dental object detection model (CranioCatch, Eskisehir, Turkey).

**Figure 3 diagnostics-14-00890-f003:**
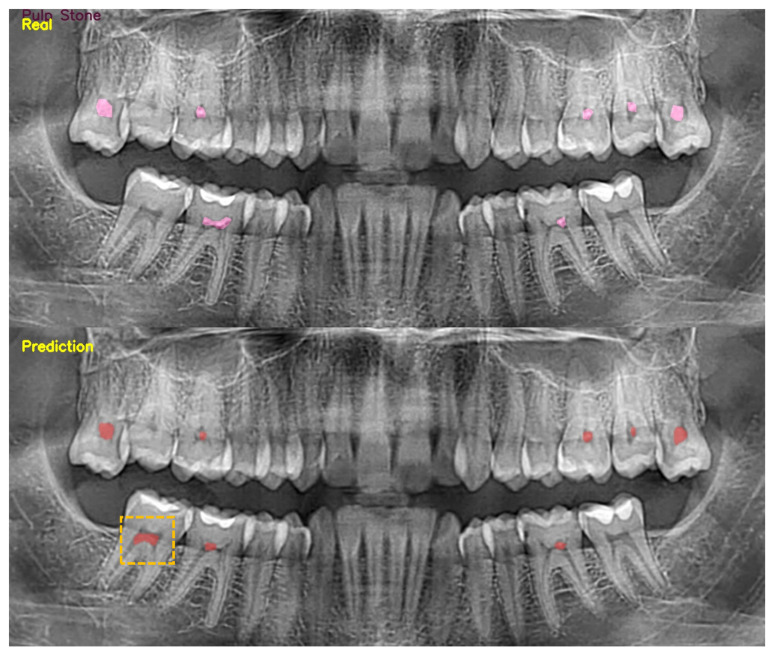
Sample images correctly and incorrectly (dashed square) predicted by neural networks on test dataset.

**Figure 4 diagnostics-14-00890-f004:**
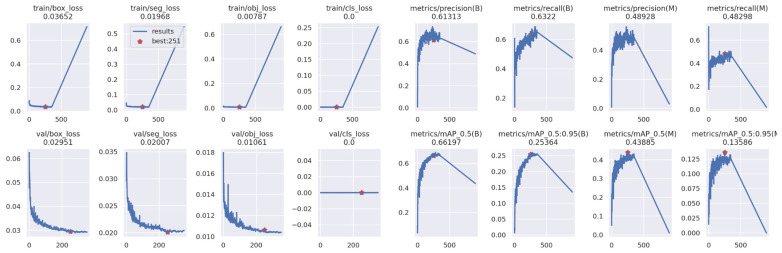
The training outcomes for detecting pulp stones utilizing the YOLOv5x algorithm.

**Figure 5 diagnostics-14-00890-f005:**
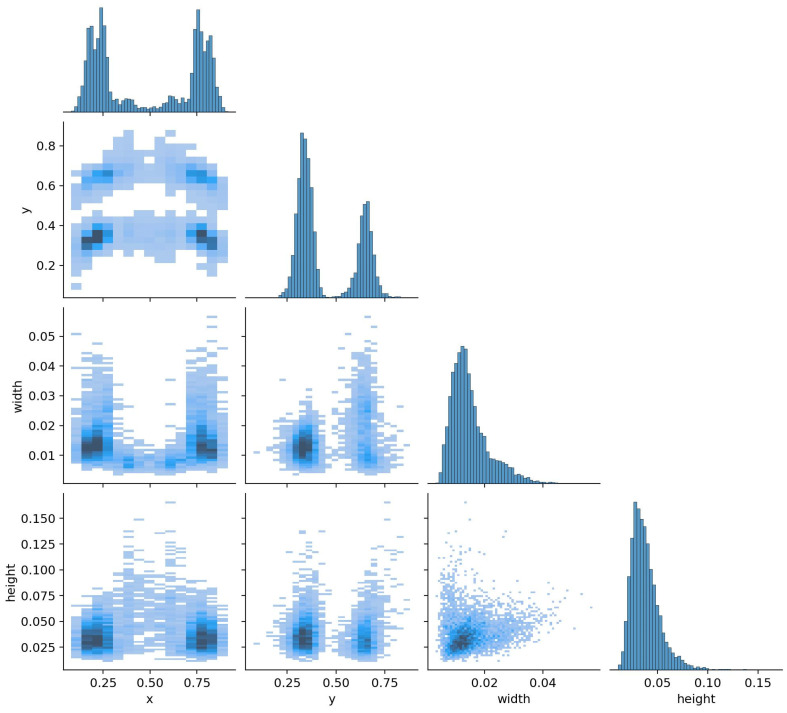
Correlogram representations for dental tooth numbering labels.

**Table 1 diagnostics-14-00890-t001:** Sensitivity, precision, and F1 results of artificial intelligence model obtained using YOLOv5 architecture.

True Positive (TP)	False Positive (FP)	False Negative (FN)	Sensitivity	Precision	F1 Score
614	177	151	0.8026	0.7762	0.7892

## Data Availability

Data is available from the corresponding author upon reasonable request.

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
