# Peer review of "The Detection of Pulp Stones with Automatic Deep Learning in Panoramic Radiographies: An AI Pilot Study"

_diagnostics, 2024, doi:10.3390/diagnostics14090890_

Round 1

Reviewer 1 Report

Comments and Suggestions for Authors

In this manuscript, the authors evaluated the efficiency of deep learning methods in the detection of pulp stones. The overall quality is good.

One small suggestion: the manuscript would be more self-sufficient if the authors could add a figure to explain the YOLOv5 model structure and have a little bit more comparison with other neural networks mentioned in the discussion section. This could be helpful in understanding the performance. 

Author Response

Thank you very much for your valuable comments.

We have revised the manuscript according to the suggestions addressed by the referee. Please find the reply letter written to the referee below. We declare that all the corrections and extensions have been written in red color. Except for the corrections and extensions suggested by the referee, nothing has been added to the paper. Besides that, nothing has been removed from the paper except the stated ones in the response. Before explaining the changes made in this revision. We thank the referee for his/her corrections and suggestions that considerably improved the paper.

Corrections and additions were made according to your recommendations.

Figure of YOLOv5 model structure added.

Discussion section improved.

Changes are marked in red.

Reviewer 2 Report

Comments and Suggestions for Authors In the considered manuscript, the authors collect and mark-up a dataset of panoramic radiography images and then apply YOLO neural network for automated detection of pulp stones. The paper is generally well-written and clearly structured.
However, I see two major (related) problems that prevent me from recommending acceptance. These are unclear contribution and no justification of novelty (see details below). The authors could probably correct these problems, so I recommend major revision.

== Contribution ==
* The authors apply a well-known YOLO architecture for a standard purpose, and do not provide any comparison to the alternatives. The absolute F1, etc. values for their dataset are of little interest if there are no baselines. Correspondingly, I do not see a research component in the manuscript or the scientific outcome. Conclusions like 232: "It is envisaged that the success rate of our deep-learning model will see improvements with the utilization of larger datasets." - are obvious and have been long known before the current work.
* In terms of architecture selection, it seems that YOLO cannot be recommended as the preferred one. In the Discussion (which has good breadth, but the specified models' quality values are not directly comparable). The authors even mention apparent Mask R-CNN's superiority - so, why should their approach be preferred?
* The dataset collected and marked up by the authors could have been a contribution, but I did not find any statements about data sharing (although it's required by MDPI, as I know). Correspondingly, it is unclear how other researchers could have used it, to validate the authors' results or for a new study.

== Novelty ==
* A related problem is that the authors do not satisfactorily explain what is original about their work. In general, application of Deep Learning methods for dental images (and particularly in panoramic ones) does not seem to be new, see e.g.
Muresan, M. P., Barbura, A. R., & Nedevschi, S. (2020, September). Teeth detection and dental problem classification in panoramic X-ray images using deep learning and image processing techniques. In 2020 IEEE 16th International Conference on Intelligent Computer Communication and Processing (ICCP) (pp. 457-463). IEEE.
* Moreover, the authors do not even explain the difference with their own previous publication, which seems to be very much related to the current manuscript (iThenticate even shows 10% similarity with it):
ALTINDAĞ, A., Sultan, U. Z. U. N., Bayrakdar, İ. Ş., & Çelik, Ö. (2023). Detecting pulp stones with automatic deep learning in bitewing radiographs: a pilot study of artificial intelligence. European Annals of Dental Sciences, 50(1), 12-16.

== Misc ==
"effectively[28,29]" - check spaces when referencing
* The body of the paper is just a bit over 3,000 words, which I think is too short for a journal paper. Possibly, the type of the manuscript could be changed from Article to a Communication. Or, considerable extension is needed.

Author Response

Thank you very much for your valuable comments.

We have revised the manuscript according to the suggestions addressed by the referee. Please find the reply letter written to the referee below. We declare that all the corrections and extensions have been written in red color. Except for the corrections and extensions suggested by the referee, nothing has been added to the paper. Besides that, nothing has been removed from the paper except the stated ones in the response. Before explaining the changes made in this revision. We thank the referee for his/her corrections and suggestions that considerably improved the paper.

Corrections and additions were made according to your recommendations.

In the literature, Mask R-CNN and YOLOv4 architectures were previously preferred in pulp stone detection studies with artificial intelligence in bitewing images. Therefore, we wanted to use a different imaging method and a different architecture in this study.

We expanded the discussion section and noted its difference from our previous work.

Changes are marked in red.

Round 2

Reviewer 2 Report

Comments and Suggestions for Authors

I have read the authors' replies to my comments and the revised version of the manuscript. According to the highlights in the manuscript and the authors' statement, the only changes are two additional paragraphs in the Discussion. Correspondingly, I do not consider that the following has been addressed:

- no contribution is clearly stated,

- no novelty is clearly explained,

- difference with own previous work not explained in the Introduction,

- the paper was not significantly extended (volume-wise, it is still below the threshold for a journal paper),

- no dataset sharing policy stated.

Moreover, 3 new references were added, and the authors use them as the baselines, which are actually vain - 2 of the references are over 5 years old, which is a huge time in ML.

Correspondingly, I still hold with major revision.

Author Response

Thank you very much for your valuable comments. We have revised the paper according to your suggestions.

Comment 1,2,3: no contribution is clearly stated, no novelty is clearly explained, difference with own previous work not explained in the Introduction.

Reply 1,2,3: Corrections were made in the introduction section in line with your suggestions.

‘’In this study, unlike previous research studies, a new effort was aimed in the field of automatic pulp stone detection. While in previous studies, Mask R-CNN and YOLOv4 architectures were primarily used to detect pulp stones in bitewing images, in this research, a new approach was presented and the effectiveness of YOLOv5, a current architecture, was evaluated in the field of panoramic images. This departure from other methodologies underscores our commitment to advancing the field by exploring innovative means of accurate and effective pulp stone detection. In this study, where YOLOv5's architecture is used in the context of panoramic radiographs, it is thought to not only expand the scope of automatic detection techniques but also provide new and different information to the existing literature on dental radiology.’’

Comment 4,6: the paper was not significantly extended (volume-wise, it is still below the threshold for a journal paper). Moreover, 3 new references were added, and the authors use them as the baselines, which are actually vain - 2 of the references are over 5 years old, which is a huge time in ML.

Reply 4,6: In the discussion section, corrections were made in line with your suggestions and expanded with current literature.

‘’Salahin et al. [2023] conducted a study focusing on the implementation and validation of an automated system for carious lesion detection from smartphone images, employing deep learning techniques. The YOLOv5X and YOLOv5M models demonstrated superior performance compared to alternative models within the same dataset. Similarly, Zhou et al. [2024] investigated the efficacy of deep learning methodologies in classifying and detecting recurrent aphthous ulcerations using clinical photographs of the oral region. Their findings indicated that ResNet50 achieved the highest success in image classification, while the pretrained YOLOv5 architecture excelled in object detection. Additionally, Ayan et al. [2024] evaluated the caries diagnosis performance of dental students following training with an artificial intelligence application. Using the YOLOv5 program, caries lesions were annotated in 1200 images by two experts, with 1000 randomly selected images utilized for student education and the remaining 200 images reserved for evaluating AI-assisted caries diagnosis. The overall class mean average precision (mAP) scores for YOLOv5s and YOLOv5xs were 63% and 65%, respectively. Additionally, the results of their study showed that students receiving artificial intelligence training in carious lesion detection demonstrated promising results.’’

Comment 5: no dataset sharing policy stated

Reply 5: I'm adding the github link below.

 https://github.com/ibrahimsevkibayrakdar/PanoramicPulpStone

Changes are marked in red.

Thank you so much for your time, effort, and contribution.

Round 3

Reviewer 2 Report

Comments and Suggestions for Authors

I have read the authors' replies and the revised version of the manuscript. I commend the authors for addressing my comments and further improving the paper.

Although I still believe that the manuscript's length is short of a journal paper, I would leave this decision to the editor. Otherwise, I do not mind the acceptance.